# Numerical Simulation of the Ice Breaking Process for Hovercraft

**Jiangjie Jin, Li Zhou \*, Shifeng Ding and Yingjie Gu**

School of Naval Architecture and Ocean Engineering, Jiangsu University of Science and Technology,
Zhenjiang 212003, China; 192010024@stu.just.edu.cn (J.J.); 201800000060@just.edu.cn (S.D.);
199010038@stu.just.edu.cn (Y.G.)
* Correspondence: zhouli209@hotmail.com

**Abstract:** A hovercraft can adapt to an ice area, open water, land and other environments, owing to its unique hull structure. It also plays an important role in transporting supplies, rescuing people, breaking ice and conducting other tasks. Ice load prediction is very important for structural safety and navigation of a polar ship, especially in design of air cushion icebreakers or ice breaking platforms. In this paper, based on a simplified circumferential icebreaking pattern, the icebreaking force of the hovercraft operating on the ice sheet at low speed is simulated in a numerical way. Numerical analysis of the icebreaking process with different ice thicknesses and bending strengths are performed. The numerical results are compared with model test data in a time domain for three operating cases. By analyzing the average ice force, the errors between numerical simulation results and model test measurements are less than 30%. The present study is significant for the preliminary design of new icebreaking hovercraft and it assists the operation possibility for existing hovercraft.

**Keywords:** hovercraft; icebreaking force; numerical simulation; model test; ice resistance

## 1. Introduction

The Arctic region has become the focus of research because of its special geographical position and resources. Usually, icebreakers are used to ensure safe navigation of ordinary vessels under severe ice conditions. Traditional polar icebreakers are large in draught, easy to open up channels in open areas, but it is difficult to break ice at shallow waters or swampy ground near shore. The hovercraft is much smaller and more flexible than a typical icebreaker, and its unique propulsion mode makes it less demanding on water depth in normal activities. Therefore, it is convenient for a hovercraft to carry out operations such as opening channels and transporting supplies in the sea.

As early as the 1970s, Canada tried to used hovercrafts in icy waters and found they had good ice-breaking capacity. During trials in the winter of 1971–1972 near Yellowknife, N.W.T., the towed hovercraft ACT-100 broke ice up to 0.7 m thick while maintaining a speed at about 5 km/h [1]. A similar satisfied performance was observed in the following winter during simulated cable ferry trials across the MacKenzie River near Tuktoyaktuk, N.W.T [1]. In subsequent experiments, researchers integrated the ACT-100 air cushion platform with an icebreaker to break ice. It demonstrated that higher efficiency was obtained for the combined operation than when only using an icebreaker under the same power. It could sail with much less ice resistance and break ice much faster. Moreover, unlike conventional icebreakers, the combination of a ship and hovercraft caused minimal damage to port facilities. For a self-propelled hovercraft, not only is it necessary to install a pad-lift fan that delivers air to the hover apron, but also to install an air propeller on the hull deck to provide additional propulsion.

The ice-breaking efficiency and mode of hovercrafts at high speed are quite different from that at low speed. Muller [2] established a numerical model for ice force calculation at low speed in theoretical analysis, and proposed an ice-breaking mechanism of hovercraft at

both high and low speeds. Hinchey et al. [1] conducted a model test of hovercraft breaking ice at low speed and analyzed corresponding ice resistance. Hinchey [3] carried out model tests and believed that the theory of gravity currents was more consistent with reality. Lu et al. [4,5] analyzed the ice-breaking mechanism of hovercraft at high speed and came up with the critical speed. They also conducted numerical simulations with LS-DYNA software. Liu et al. [6,7] used the combination of the Boundary Element Method (BEM) and Finite Element Method (FEM) to perform a numerical simulation. However, FEM is inefficient in its calculation. Its numerical simulation results depend heavily on the size of the finite element block. Therefore, other more effective numerical solutions need to be considered.

Based on discretization of the ice field with square, a mechanical model of the interaction between ship hull and sea ice is established to simulate the icebreaking process of hovercraft at low speed in this paper. The time history curve of ice load acting on the ship hull is obtained. The numerical simulation results are compared with the published model test results. Finally, the parameter analysis of the numerical simulation is carried out.

## 2. Method Model

### 2.1. Ice Breaking Mechanism

The icebreaking mode of hovercraft can be divided into two types: low speed and high speed, which are also known as static icebreaking and dynamic icebreaking. In the low speed mode, high pressure gas will act on the water surface, displace the nearby water, create a depressed circular surface and radiate to the bottom of nearby ice sheet. As a hovercraft approaches ice sheet, an air cavity forms beneath the ice sheet, creating a cantilever beam effect. The length of the air cavity reaches its maximum when the apron of the hovercraft starts to contact the ice surface. The contact area increases as the ship moves forward. When resultant force acting on the ice sheet reaches the ice-bearing limit, then level ice will break and slide around the hull. Therefore, the basic principle of icebreaking in the low speed mode is to replace the water below the ice by air cavity, forming a cantilever beam effect that makes it easier to be fractured and broken down. The general principle is shown in Figure 1. F is the joint force of the ship on the ice block, Fc is the upward supporting force of high pressure gas on ice block and G is the gravity of ice block.

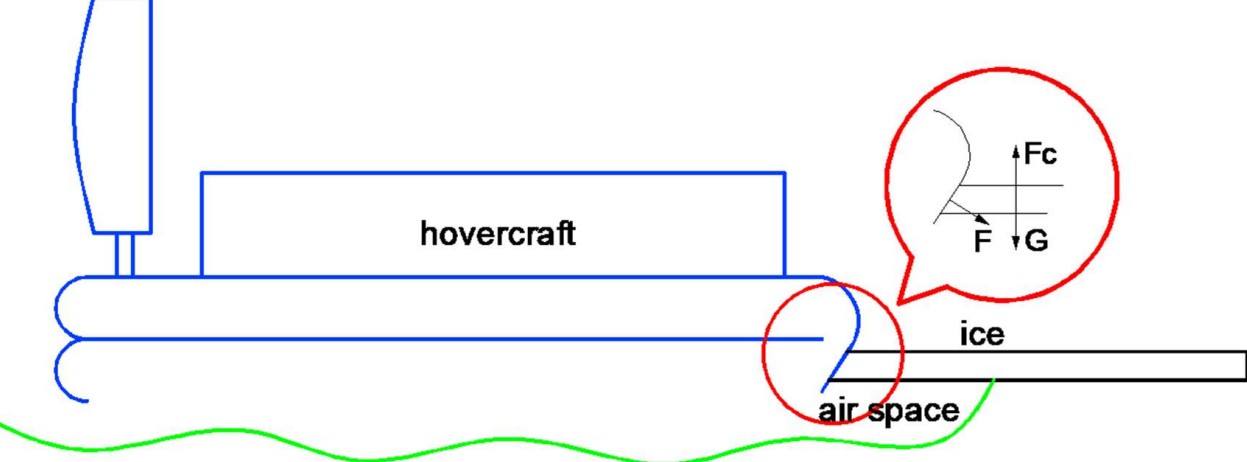

**Figure 1.** Ice breaking mechanism.

### 2.2. Ship-Ice Mathematical Model

As a hovercraft advances, the contact area between hover apron and sea ice gradually increases, and crushing force acting on the ice also increases. The crushing force ($F_{cr}$) calculation formula is expressed as [8]

$$F_{cr} = \sigma_c \times A_c \tag{1}$$

where $\sigma_c$ is the crushing strength of ice; $A_c$ is the contact area, as shown in the Figure 2.

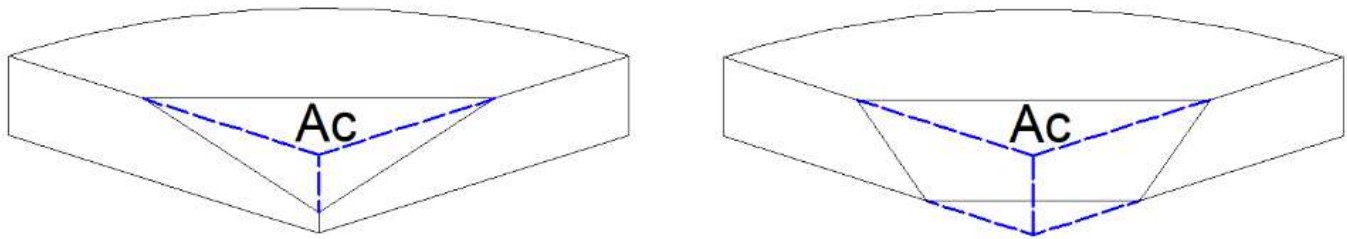

**Figure 2.** Contact area.

As shown in the Figure 3, the relative velocity during the advance of the hovercraft ship is denoted by $v^{rel}$, and it can be divided into $v_n^{rel}$ and $v_\tau^{rel}$. The frictional coefficient between the ice and the hull is denoted by $\mu$. The frictional force between the ship and the ice is denoted by $f$, and it can be divided into $f_H$ and $f_V$. The resultant force of frictional force $f_V$ and crushing force $F_{cr}$ can be divided into $F_H$ and $F_V$ in horizontal and vertical directions. Assuming that there is no vertical displacement before bending failure. So, $f_H$ is proportional to the relative velocity component $v_\tau^{rel}$ and $f_V$ is proportional to the relative velocity component $v_{n,1}^{rel}$ [8]. Furthermore, they can be expressed by Equations (2)–(5):

$$f_H = \mu F_{cr} v_\tau^{rel} / \sqrt{\left(v_\tau^{rel}\right)^2 + \left(v_{n,1}^{rel}\right)^2} \tag{2}$$

$$f_V = \mu F_{cr} v_{n,1}^{rel} / \sqrt{\left(v_\tau^{rel}\right)^2 + \left(v_{n,1}^{rel}\right)^2} \tag{3}$$

$$F_H = F_{cr} \sin\varphi + f_V \cos\varphi \tag{4}$$

$$F_V = F_{cr} \cos\varphi - f_V \sin\varphi \tag{5}$$

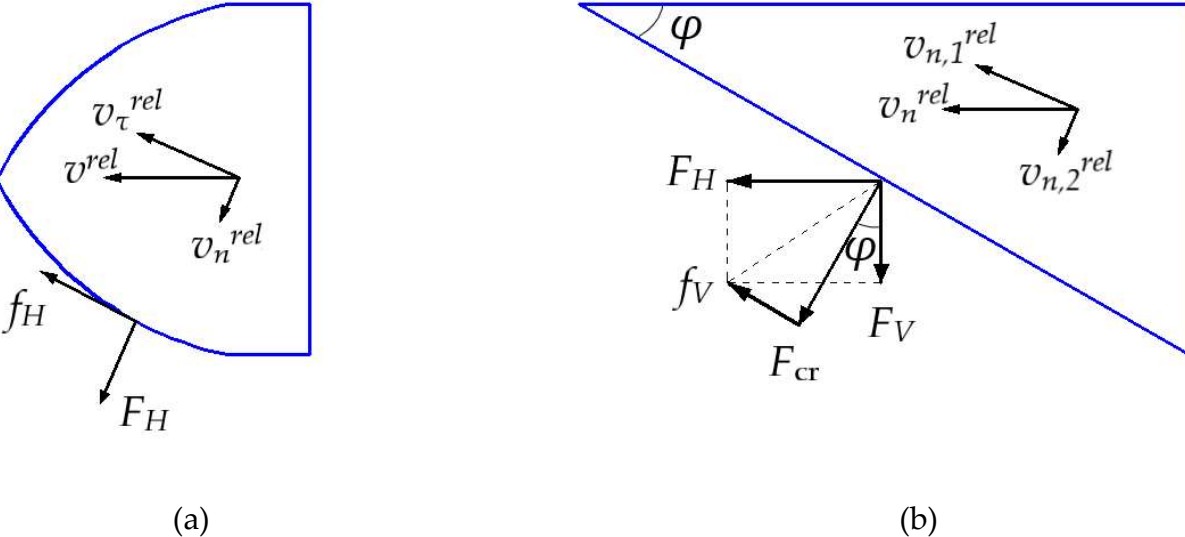

(a)

(b)

**Figure 3.** Definition of ice force and velocity components. (**a**) horizontal plane; (**b**) vertical plane.

The high pressure gas induced by the hovercraft pushes away the water under the ice sheet to form an air cavity. At this point, the ice sheet is also supported by its own gravity and the upward force of the gas. When the ice breaks, an ice wedge with an opening angle of $\theta$ is formed. Furthermore, its gravitational force ($G$) can be calculated by

$$G = \pi R^2 h_i g \times \frac{\theta}{2\pi} \tag{6}$$

where $hi$ is the ice thickness; $R$ is the radius of broken ice, g is the acceleration of gravity and $\theta$ is the opening angle of ice wedge, $\pi$ is Pi.

When the high pressure gas acts on the water surface, it will form a cavity. The depth ($d$) of the water surface depression is highly related to the air cushion pressure of the hovercraft, which is given as

$$d = \frac{P}{\rho_w g} \tag{7}$$

where $P$ is the air cushion pressure; $\rho_w$ is the density of water.

In addition, the thickness of the ice under the water surface is around 9 times that of ice above the water surface. Therefore, when d is less than 0.9 hi, the air cavity cannot be formed under the ice sheet. In Hinchey's [3] experiment, he observed that the air cavity radiates out in a semicircular shape under the ice sheet. Without considering the air leakage, the relationship between the radius and height of the air cavity is written as:

$$R' = \sqrt{2gh'(1 - \kappa)} \tag{8}$$

where $R'$ is the radius of the air cavity; $h'$ is the height of the air cavity; $\kappa$ is a constant determined by experiment, $\kappa = 0.35$.

According to the experimental results of Hinchey, the pressure in the air cavity is calculated by:

$$P' = rgh'(1 - \kappa) \tag{9}$$

where $P'$ is the air cavity pressure; $r$ is the air cushion radius of the hovercraft.

Therefore, when the radius of the air cavity is greater than or equal to the icebreaking radius, the Fc can be expressed numerically as:

$$F_c = P'A' = \frac{R'^2 r}{2} \tag{10}$$

where $A'$ is the area of broken ice.

### 2.3. Ice Failure Criterion

There are two main failure patterns of sea ice which include crushing and bending failures. When a ship moves against sea ice, ice failure patterns are complicated. Single ice failure pattern or a mixture of two ice failure patterns might occur during the interaction. The failure patterns of sea ice are affected by many factors, including slope angle of the structure, ice thickness, width of the structure, relative speed of the ice-structure and so on.

According to Zhou et al. [9], the failure patterns of ice vary under different hull slope angle, which is the inclined angle of the hovercraft apron in this paper. The friction coefficient shows influence on the limit slope angle. It is assumed that the friction coefficient is in the range 0.2–0.35, the minimum boundary angle that distinguish between bending failure and crushing failure is 72° [9–11]. It means that the hull angle less than 72° will lead to bending failure when the ice failure condition is met. Once the ice-structure friction coefficient is known, the limitation of slope angle is determined. In this paper, the slope angle of the air cushion apron is 45°, which means that only bending failure occurs.

During the interaction process, local ice will expose both vertical compression and horizontal tension simultaneously on the hull when the hovercraft sails in the polar area. Then, ice fails in bending and crushing at most occasions. Circumferential cracks parallel to

the contact surface or radial cracks perpendicular to the contact surface will emerge under the effect of compression and tension. The assumption adopted in this paper is ice block breaks from ice sheet in the vertical direction and its contact surface is flat. The contact area can be determined by the length and depth of contact area. The geometric shape of the ice floes broken from level ice can be assumed to be wedge-shaped. The angle of ice wedge is θ, and the ice wedge is shaped based on the icebreaking radius (R), the expression of which is given in reference [12]:

$$R = C_l \times l \left( 1.0 + C_v \times v_n^{rel} \right) \tag{11}$$

where $C_l$ and $C_v$ are empirical parameters, $C_l$ is the length coefficient, $v_n^{rel}$ is the relative normal velocity between ice and hull and $l$ is the characteristic length of ice:

$$l = \left[ \frac{E_i h_i^3}{12(1 - v)^2 \rho_w g} \right]^{1/4} \tag{12}$$

where $E_i$ is the young's modulus of ice, $hi$ is the thickness of ice and $v$ is the Poisson ratio of ice.

The ice wedge in contact with the hull is under the ice load in vertical direction. Then the bearing capacity of ice ($P_f$) can be introduced based on the reference [9]:

$$P_f = C_f \times \left( \frac{\theta}{\pi} \right)^2 \times \sigma_f \times h_i^2 \tag{13}$$

where $\sigma_f$ is the bending strength of ice and $C_f$ is the fracture coefficient and an empirical parameter.

### 2.4. Ship Motion Equation

We assume that the Z vertically upward, X in the direction of forward motion, and the origin at the hull's center of gravity. Then, linear coupled differential equations of motion can be written as:

$$(M + A)\ddot{r}(t) + B\dot{r}(t) + Cr(t) = F \tag{14}$$

If the ship is lateral symmetry:

$$M = \begin{bmatrix} M & 0 & 0 \\ 0 & M & 0 \\ 0 & 0 & I_{66} \end{bmatrix} \tag{15}$$

$$A = \begin{bmatrix} A_{11} & 0 & 0 \\ 0 & A_{22} & A_{26} \\ 0 & A_{62} & A_{66} \end{bmatrix} \tag{16}$$

where $M$ is the mass of the ship, $I_{66}$ is the moment of inertia in Z direction and $A$ is the added mass.

The damping coefficient (B) and hydrostatic restoring coefficient (C) are assumed to be zero in this 3DOF (3 Degrees of Freedom) mode.

### 2.5. Ice Force Model

Ice resistance includes the continuous icebreaking forces $F_1^{brk}(t)$ and ice submerging forces. The local icebreaking forces change under the different parts of the hull at each time step. The total continuous icebreaking forces are calculated based on the local icebreaking

forces. The ice submerging forces are calculated according to the ice resistance formula, then the ice resistance in general can be expressed as [8]:

$$F_1^{ice}(t) = F_1^{brk}(t) + R_s \left( 1 + 9.4 \frac{v^{rel}}{\sqrt{gL_{WL}}} \right) \times \frac{v_1^{rel}}{v^{rel}} \tag{17}$$

$$F_2^{ice}(t) = F_2^{brk}(t) + R_s \left( 1 + 9.4 \frac{v^{rel}}{\sqrt{gL_{WL}}} \right) \times \frac{v_2^{rel}}{v^{rel}} \tag{18}$$

$$F_6^{ice}(t) = F_6^{brk}(t) \tag{19}$$

where $v^{rel}$ is the relative velocity between ice and hull, $v_1^{rel}$ is the forward component of $v^{rel}$, $v_2^{rel}$ is the transverse components of $v^{rel}$; $L_{WL}$ is the water line length of the ship; $R_s$ is the submersion component of ice resistance, which can be written as [13]:

$$R_S = (\rho_w - \rho_i)gh_iB \left( T \frac{B+T}{B+2T} \right) + \mu \left[ \left( 0.7L - \frac{T}{tan\varphi} \right) - \frac{B}{4tan\alpha} \right] + Tcos\varphi \cdot cos\psi \sqrt{\frac{1}{sin^2\varphi} + \frac{1}{tan^2\alpha}} \tag{20}$$

where $\rho_w$ is the density of sea; $\rho_i$ is the density of the level ice; $B$ is the beam; $T$ is the draught; $\varphi$ is the stem angle of the ship; $\alpha$ is the water entrance angle, $\psi$ is the angle between the normal and vertical directions of the hull surface, $\psi = $ arc tan $(tan\varphi/sin\alpha)$.

## 3. Experiment Description and Numerical Validation

### 3.1. Experiment Description

In this paper, the related model tests of a hovercraft in an ice basin were carried out in the State Key Lab of Hydraulic Engineering Simulation and Safety of Tianjin University [14]. The measurements were used to validate the simulation results by the present method. The ice basin is 40 m long, 6 m wide and 1.8 m deep. It could make ice sheet with thickness of 1~30 cm. The laboratory area under cooling is 320.0 m$^2$, and the temperature in the laboratory can be adjusted from $-25\,°C$ to $0\,°C$.

This model test is the towing test of structures in the ice area. The model test follows the Froude similarity and the Cauchy similarity. A model of large air cushion platform was used in the tests. The model scale ratio is 1:5. The main parameters of the platform in both full and model scales are shown in Table 1 [15]. The hovercraft model is shown in Figure 4. The model was connected to the trailer through a force sensor and pulled through the ice field. This sensor was fixed in the middle of the bow of the hovercraft. The sampling frequency of the sensor is 100 Hz, and the accuracy is 5% [15].

**Table 1.** Main parameters of the ship.

| Items | Full Scale | Model Scale |
|---|---|---|
| Length Overall/m | 11.95 | 2.39 |
| Cushion Length/m | 13.55 | 2.71 |
| Breadth/m | 6.9 | 1.38 |
| Cushion Breadth/m | 8.45 | 1.69 |
| Cushion Height/m | 0.8 | 0.16 |
| Air Pressure/Pa | 2642 | 534.8 |
| Speed/kn | 1 | 0.447 |

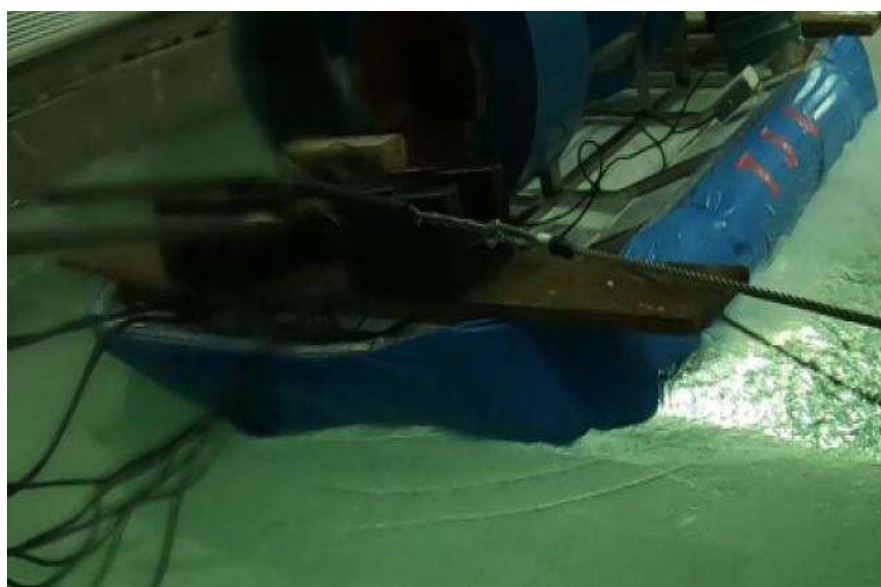

**Figure 4.** Hovercraft model used in the model test [15].

### 3.2. Ice Model

Urea ice was used in this experiment, which can increase the porosity and reduce the strength of ice. The urea ice model is shown in Figure 5. The ice crystals and growth process of urea ice are similar to first-year ice in the Arctic. Therefore, it can be ensured that the key characteristics of urea ice are highly similar to the Arctic sea ice, such as failure pattern and strength characteristics.

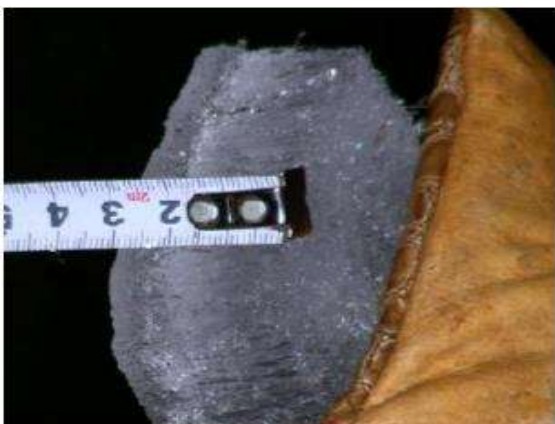

**Figure 5.** Ice made in the basin [15].

When the model ice thickness grows to expected value, the strength of the ice is controlled by adjusting the temperature. By using the cantilever or simple beam approach, the bending strength could be measured. When the average value of model ice strength reaches the predefined value approximately, the ice sheet is ready for the tests. There are three cases in the tests, and the ice parameters from measurements are shown in Table 2.

**Table 2.** Main parameters.

|  | Case | Towing Speed/kn | Ice Thickness/mm | Bending Strength/kPa |
|---|---|---|---|---|
| 1 | Model scale | 0.447 | 32 | 43.7 |
|  | Full scale | 1 | 160 | 218.5 |
| 2 | Model scale | 0.447 | 58 | 48.6 |
|  | Full scale | 1 | 290 | 243 |
| 3 | Model scale | 0.447 | 73 | 62 |
|  | Full scale | 1 | 365 | 310 |

### 3.3. Numerical Simulation

In this paper, the full-scale hovercraft parameters ware used for three cases in numerical simulation. The main hovercraft parameters are shown in Table 3.

**Table 3.** Ship and ice characteristics.

| Items | Value |
|---|---|
| Length Overall/m | 13.55 |
| Breadth/m | 8.45 |
| skirt height/m | 0.8 |
| Air Pressure/Pa | 2642 |
| Ice thickness/m | 0.16/0.29/0.365 |
| Bending strength/kPa | 218.5/243/310 |
| Speed/kn | 1 |
| Ice density/kg/m$^3$ | 900 |
| Water density/kg/m$^3$ | 999.8 |

An ice field with a certain size is predefined according to the dimension of the ship. Then, we divide the level ice into a number of square blocks, as shown in Figure 6a. The length of each ice grid could be taken as icebreaking length, which is calculated as Equation (11) [16]. The waterline of the hovercraft is assumed to keep constant in the simulation. After fixing the draught of the hovercraft, the waterline is divided into countless discrete nodes. The bow and stern of the hull are taken as circular approximately. The initial bird view of ice–hull interactions is presented in in Figure 6b.

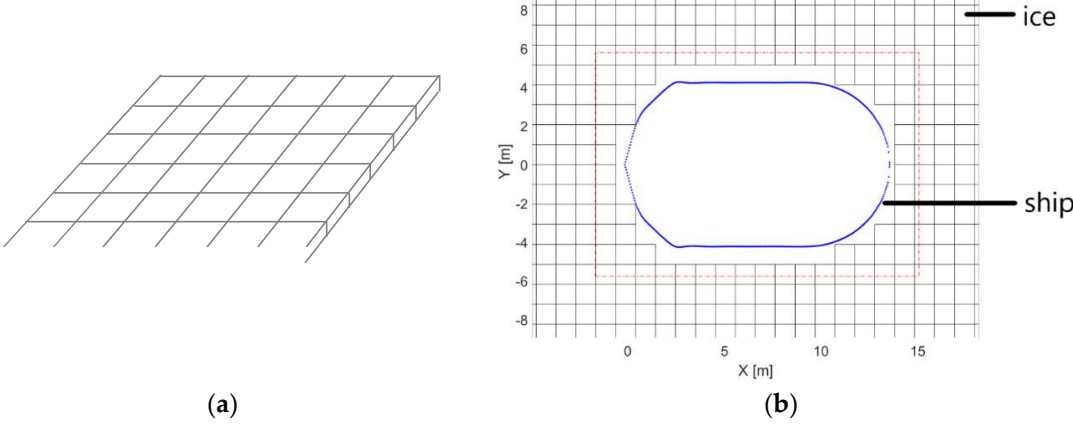

(**a**)             (**b**)

**Figure 6.** Sketch of the ice and the ship. (**a**) Ice field discretion; (**b**) Initial location of ship in ice.

In order to improve computation efficiency, we set a rectangular area in red around the hull node, as shown in Figure 6. Only ice nodes, which are in the red region, need to be considered to upgrade or not. Each ice grid in the region needs to be checked if it interacts with the hull or not. Once the resulting ice force exceeds the bearing capacity, the ice

boundary should be updated. Then, the position of ship is updated at each time step based on the results solved by the equation of motion of the previous time step. Finally, an intact ship sailing trail and time history curves of the ice force are obtained from the simulation.

In the experiment, it is observed that forward speed of the air cavity was greater than the ship's forward speed, and two air cavities would merge when the hovercraft moved straightly. Therefore, in the numerical simulation, it is assumed that the length of the air cavity is greater than the length of the ice block. Then we performed numerical simulations with three cases, and ice parameters are shown in Table 2. The simulated time history curves of ice force in longitudinal direction for three cases are shown in Figures 7–9. The ice load is oscillating periodically at stable state. The peaks are not constant and vary a little bit.

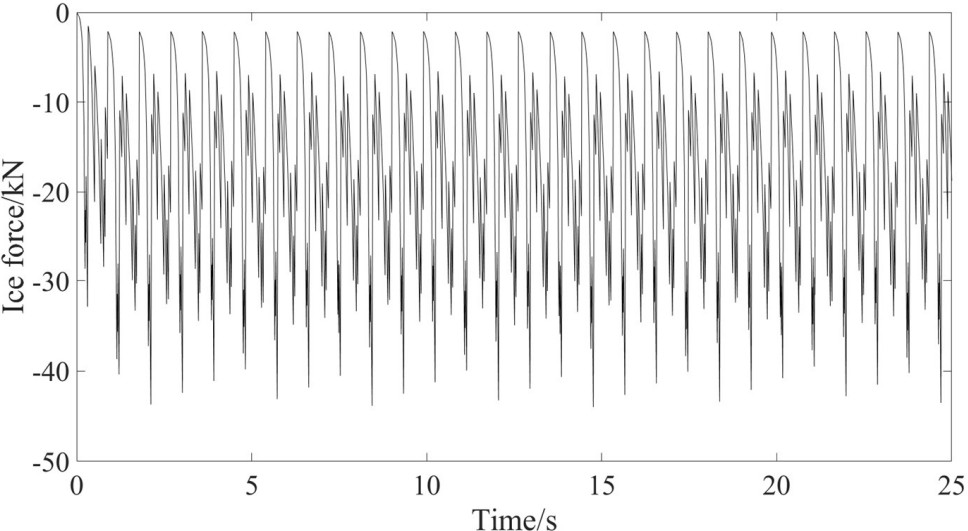

**Figure 7.** Time history curves of total ice force in case 1.

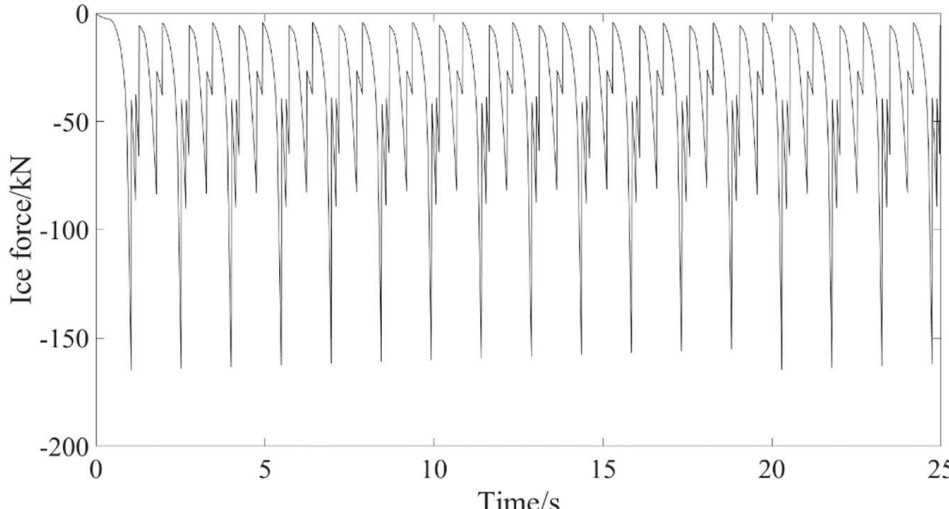

**Figure 8.** Time history curves of total ice force in case 2.

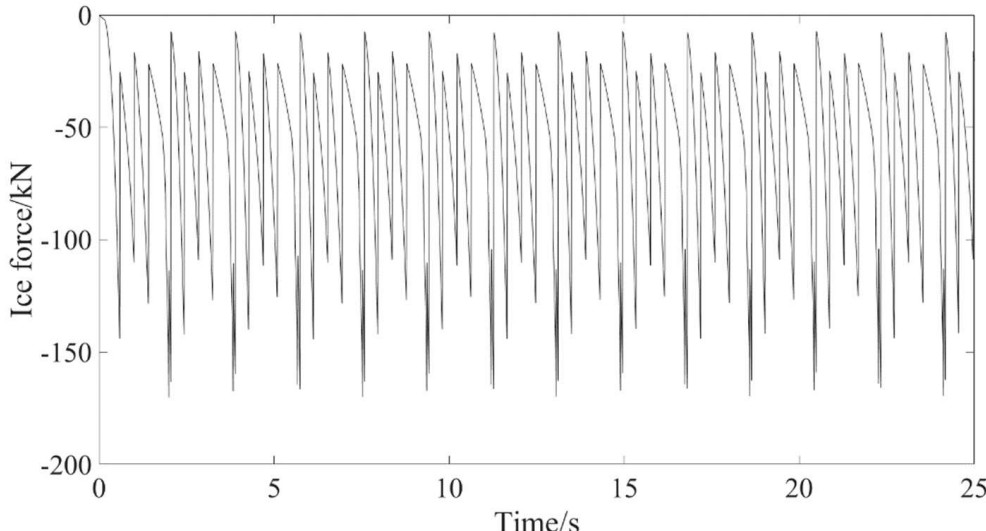

**Figure 9.** Time history curves of total ice force in case 3.

The comparison between the numerical simulation results and experimental results is shown in Table 4. We can get some conclusions by analyzing the numerical simulation results and the experimental results. In case one, the average ice force of numerical simulation is 16.73 kN, the experimental result is 23.73 kN and the relative error is 29.85%. The maximum ice force of numerical simulation is 43.98 kN, the experimental result is 132.25 kN and the margin of error is 66.7%. In case two, the average ice force of numerical simulation is 37.89 kN, the experimental result is 41.25 kN and the margin of error is 8.15%. The maximum ice force of numerical simulation is 165.10 kN, the experimental result is 226.38 kN and margin of error is 26.9%. In case three, the average total ice force of numerical simulation is 61.63 kN, the experimental result is 65.63 kN and the margin of error is 6.09%. The maximum ice force of numerical simulation is 170.20 kN, the experimental result is 265.03 kN and the margin of error is 35.8%. As shown in Figures 7–9, force shows a periodic oscillation trend, which is similar to the trend of force measured in the experiment. This is because when hull contacts the ice, force will have a loading trend, and when force reaches the maximum bearing capacity of ice, ice will break and then force will have a unloading trend, which is also consistent with the phenomenon observed in the experiment. As the ship moves forward, the loading and unloading process of force occurs alternately and force show an oscillation trend in the time history curve.

**Table 4.** Comparison of simulated and measured ice force.

| Case | Average in Simulation/kN | Average in Experiment/kN | Error/% | Maximum in Simulation/kN | Maximum in Experiment/kN | Error/% |
|---|---|---|---|---|---|---|
| 1 | 16.73 | 23.73 | 29.58 | 43.98 | 132.25 | 66.7 |
| 2 | 37.89 | 41.25 | 8.15 | 165.10 | 226.38 | 26.9 |
| 3 | 61.63 | 65.63 | 6.09 | 170.20 | 265.03 | 35.8 |

In the numerical simulation results, the margin of error is close to 30% in case one. The numerical simulation results of the other two cases are better, and both margins of error are less than 10%. The margin of error in case two is 8.15% and for case three it is 6.09%.

The first reason may be the experiment itself; each case was only carried out once. The measured data exist with some uncertainties; waves created by high-pressure gas and the accuracy of the sensor will lead to such uncertainties. The second reason may be the setting of ice parameters in the numerical simulation. In the experiment, some parameters were not clearly measured, such as crushing strength, Young's modulus, friction coefficient, et al. Therefore, in the numerical simulation, these inputs can only be set based on general laws and with reference to real sea ice. For the error of maximum ice force, we do not consider the case, where gas leakage during the contact between the hovercraft and ice. In ship trails,

when the ice is broken, gas will be ejected upwards along the cracks, and the hovercraft will be slightly tilted upward. At this time there will be a resistance; we call it gas leakage resistance. However, we did not consider it in the simulation. According to the phenomena described in the model tests and the recorded force, the peak force occurred at the time when the hovercraft first contacted with the ice. For the large error in case 1, it is considered herein that ice buckling failure and other ice failure patterns may occur when ice thickness and bending strength are relatively small, which does not match with the bending failure assumed in this paper.

## 4. Sensitivity Analysis of Parameters

When calculating ice force, the influences of bending strength, crushing strength, Young's modulus, length coefficient $C_l$, fracture coefficient $C_f$ and other parameters are relatively large, so sensitivity analysis of these parameters will be discussed. The ice parameters of case 3 in Table 4 are mainly used for sensitivity analysis. Only influence factors will change while keeping other factors constant if not stated otherwise in all simulated cases as follows.

### 4.1. Length Coefficient

Length coefficient is empirical which is often measured and observed from model test in ice basin or sea trials in ice-covered waters. In the present model, it is directly related to the length of the squared ice grid. Therefore, it should be studied. Four length coefficients as 0.21, 0.23, 0.25 and 0.27 are taken for parameter sensitivity analysis, and the time history curves of ice force are shown in Figure 10. It is clear that ice force changes rapidly and shows strong nonlinearity.

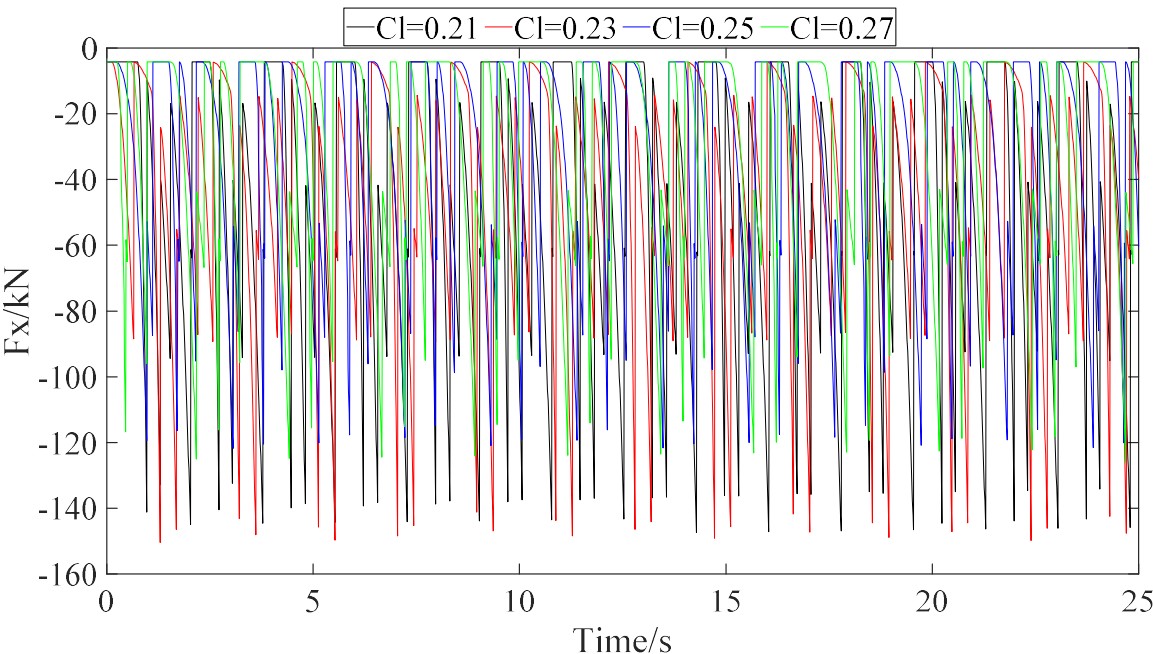

**Figure 10.** Time history curves of total ice force under different length coefficients ($C_l$).

Fast Fourier transform (FFT) is applied to get the corresponding frequency domain for simulated curves in Figure 10. The results are shown in Figure 11. It can be seen from the figure that large peaks all appear in the low frequency region, and the frequency difference between the peaks is basically equal. The reason for this situation is that the ice field has been pre-processed into square ice blocks. During the advance of the hovercraft, the ice blocks break periodically. Therefore, the frequency of peaks in this program is related to the length of the ice block. The ice length coefficient affects the length of the ice block.

Therefore, in the frequency domain curves of different length coefficients, the frequency difference between the peaks is large.

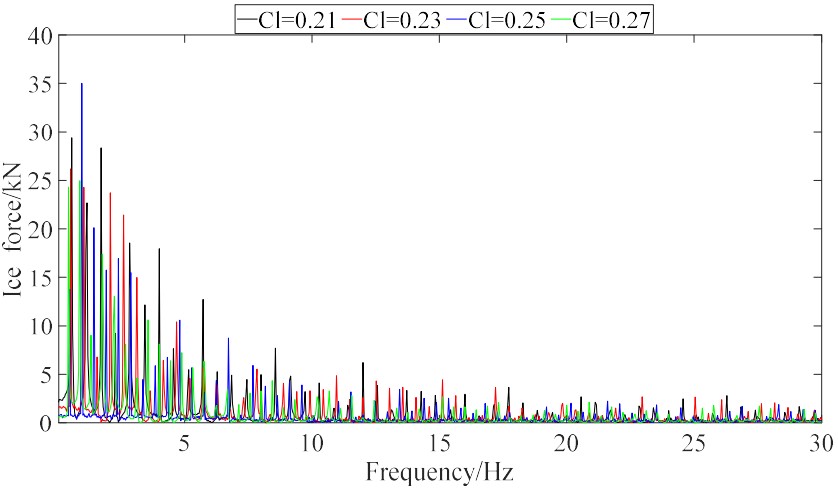

**Figure 11.** Frequency domain curves with different length coefficients ($C_l$).

In Table 5, statistics such as average ice force, maximum ice force and variance under different length coefficients are presented. These data are also plotted in Figure 12. By analyzing the curves under different length coefficients, it is clear that from the figures that the average ice force will decrease with the increase of the length coefficient. The change of length coefficient has a minor effect on the average ice force. The length coefficient has a greater influence on the variance, and the trend tends to be gentle.

**Table 5.** Statistics of ice force under different length coefficients ($C_l$).

| Length Coefficient | Average Ice Force/kN | Maximum Ice Force/kN | Variance/kN | The Frequencies of the First Four Peaks/Hz |
|---|---|---|---|---|
| 0.21 | 47.13 | 147.6 | $1.63 \times 10^3$ | 0.56/1.16/1.72/2.28 |
| 0.23 | 43.58 | 150.4 | $1.17 \times 10^3$ | 0.52/1.04/1.56/2.08 |
| 0.25 | 31.49 | 121.8 | $1.05 \times 10^3$ | 0.48/0.96/1.44/1.92 |
| 0.27 | 29.47 | 125.1 | $1.02 \times 10^3$ | 0.44/0.88/1.32/1.76 |

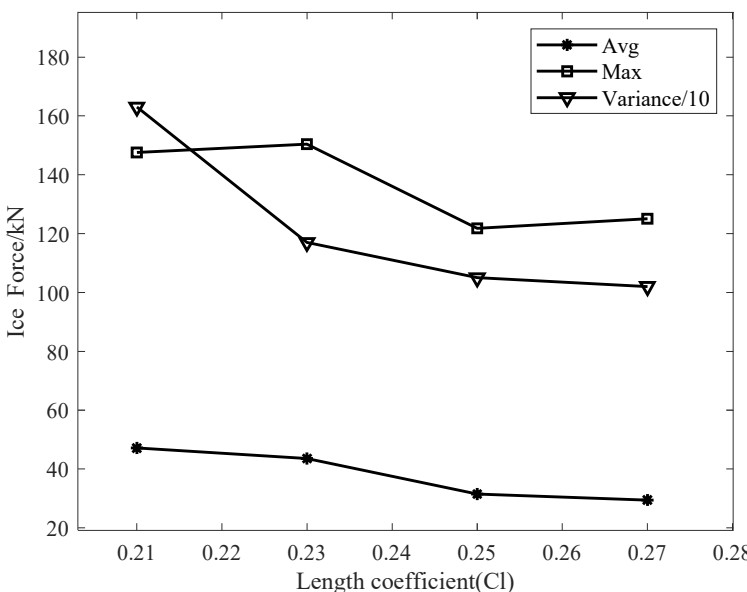

**Figure 12.** Change trend of variance and ice force with length coefficients ($C_l$).

### 4.2. Fracture Coefficient

Fracture coefficient is highly related to ice bearing capacity in the present model and may have influence on resulting ice force. Take fracture coefficients as 3.2, 3.4, 3.6 and 3.8, and time history curves of total ice force are shown in Figure 13. It can be seen from the time history curves that the number of peaks appearing in the same time is similar, which shows that the fracture coefficient has little effect on the ice breaking efficiency, but may influence the ice load.

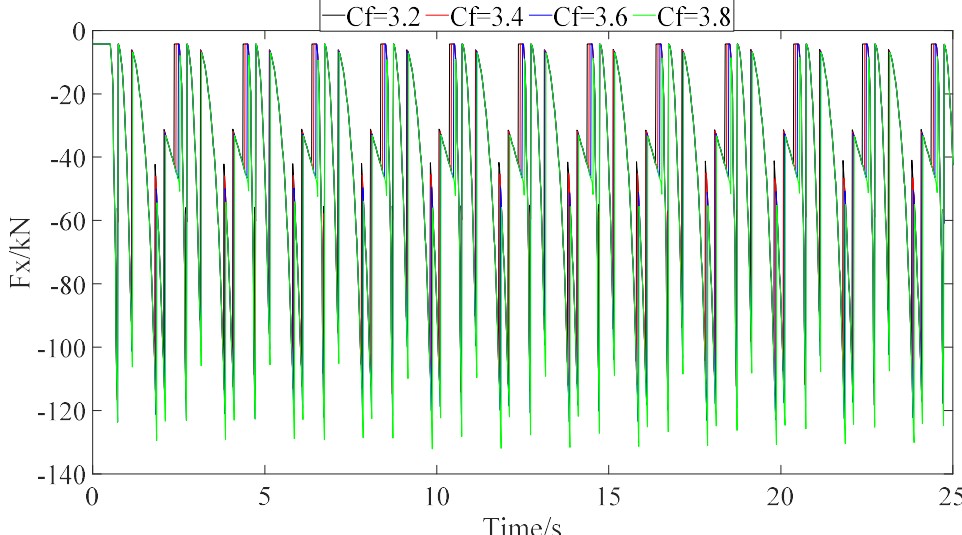

**Figure 13.** Time history curves of total ice force under different fracture coefficients ($C_f$).

Based on FFT, the frequency domain curves are shown in Figure 14. It is found that the frequencies of the peaks are all coincident. This shows that the changes in crushing strength will not affect the size of the ice block. This is because the size of the ice block is determined by Equation (11), and is only related to the length coefficient, ice thickness and Young's modulus.

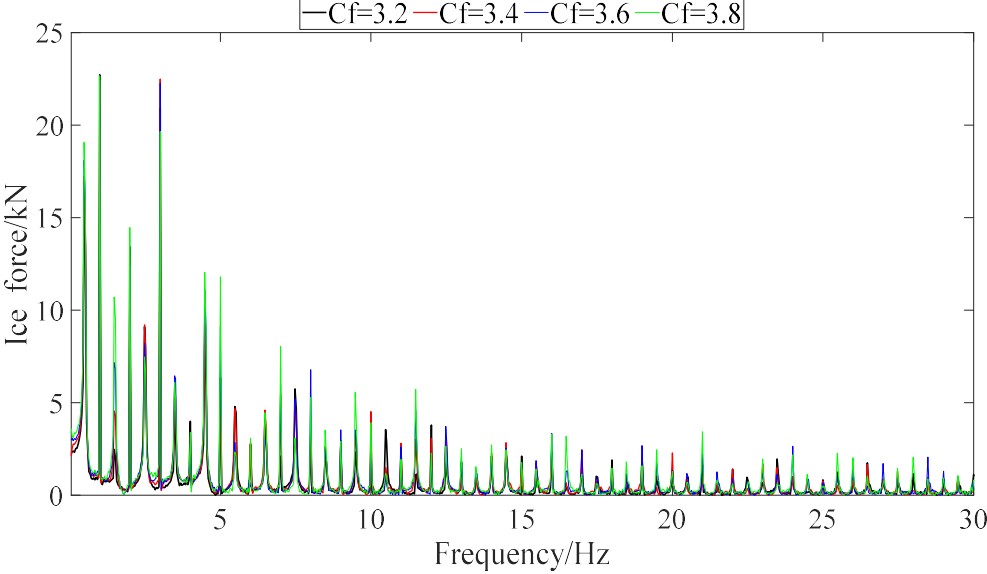

**Figure 14.** Frequency domain curves with different fracture coefficients ($C_f$).

Based on the simulated curves of ice force in Figure 13, the average ice force, maximum ice force and variance as a factor of fracture coefficient are summarized in Table 6. The simulated data are presented in Figure 15. It is found that the fracture coefficient has positive effect on average ice force, maximum ice force and variance. As the fracture coefficient increases, the average, maximum and variance ascend linearly. Compared to the average and maximum, the slope of the variance curve is large, which means that oscillation of the ship will increase.

**Table 6.** Statistics of ice force under different fracture coefficient ($C_f$).

| Fracture Coefficient | Average Ice Force/kN | Maximum Ice Force/kN | Variance/kN | The Frequencies of the First Four Peaks/Hz |
|---|---|---|---|---|
| 3.2 | 38.72 | 112.8 | $0.82 \times 10^3$ | 0.48/1.00/1.48/2.00 |
| 3.4 | 41.94 | 118.3 | $0.91 \times 10^3$ | 0.48/1.00/1.48/2.00 |
| 3.6 | 45.04 | 123.7 | $1.01 \times 10^3$ | 0.48/1.00/1.48/2.00 |
| 3.8 | 48.94 | 132.2 | $1.10 \times 10^3$ | 0.48/1.00/1.48/2.00 |

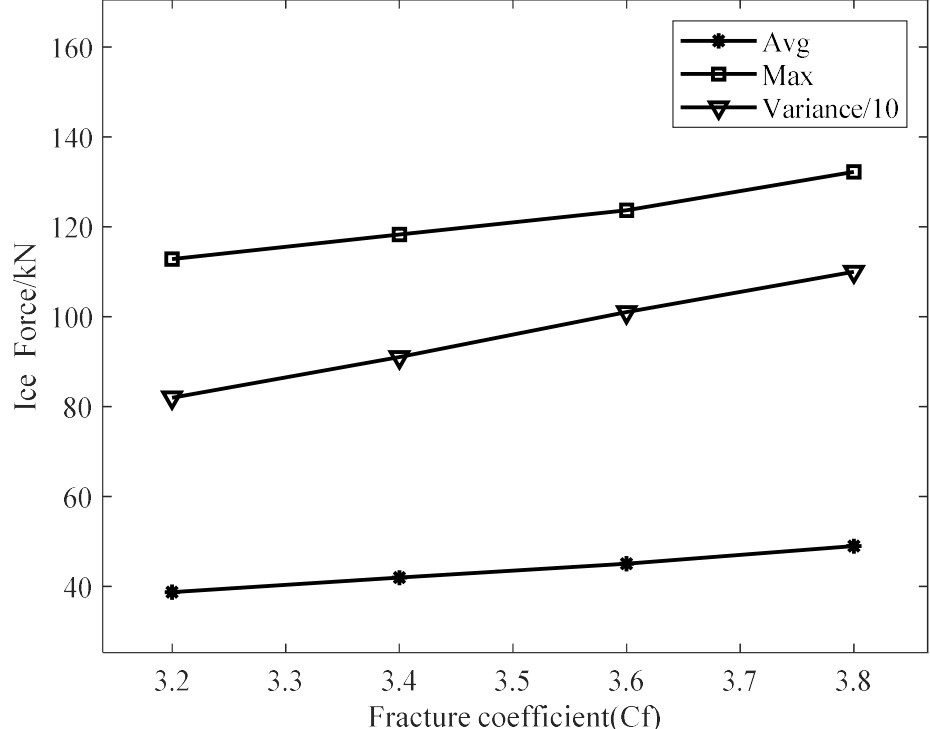

**Figure 15.** Change trend of variance and ice force with fracture coefficient ($C_f$).

### 4.3. Ice Thickness

Ice thickness is used to calculate maximum ice failure force and will affect the statistics of simulated ice force. Ice thickness varies from 0.2 m to 0.5 m according to icebreaking capability of some typical hovercrafts. The time history curves of total ice force from the simulation are shown in Figure 16. It can be seen from the time history curves that the icebreaking interval gradually increases as the ice thickness increases. This indicates that the icebreaking efficiency decreases with increasing ice thickness, based on the same energy consumption. The curves are transformed to get the corresponding frequency domain curves by FFT. The results are presented in Figure 17. In the frequency domain curves of different ice thicknesses, the frequency difference between the peaks is large. The difference between the peaks of ice thickness is greater than that of length coefficient. It means that ice thickness has the greatest effect on the length of the ice block.

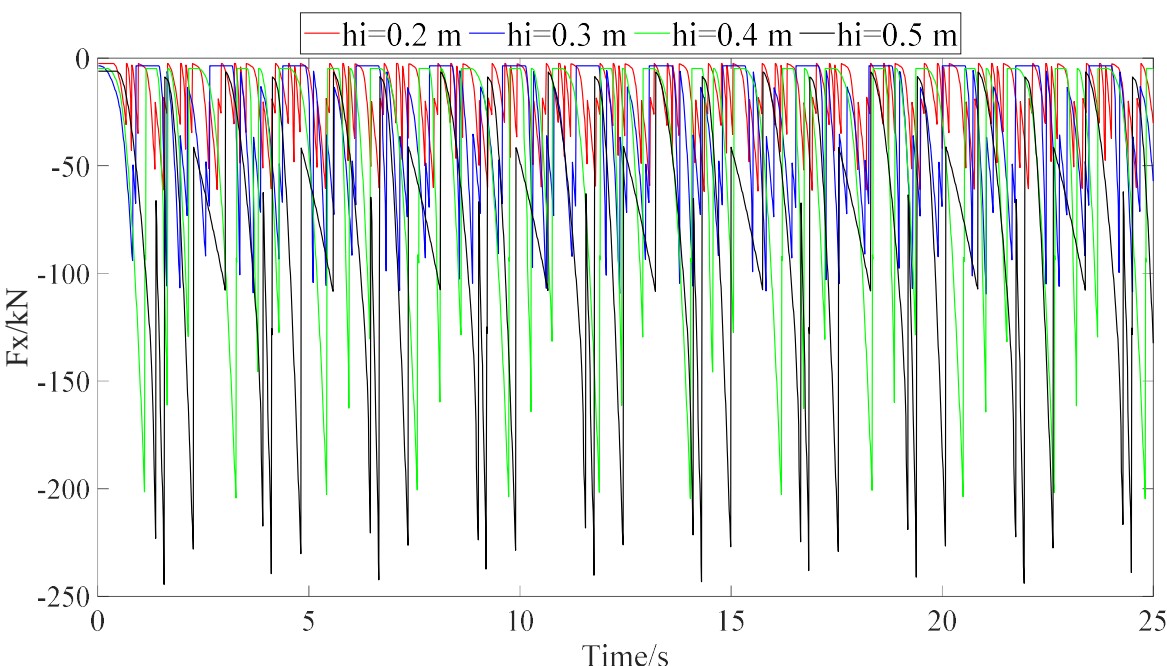

**Figure 16.** Time history curves of total ice force under different ice thickness (hi).

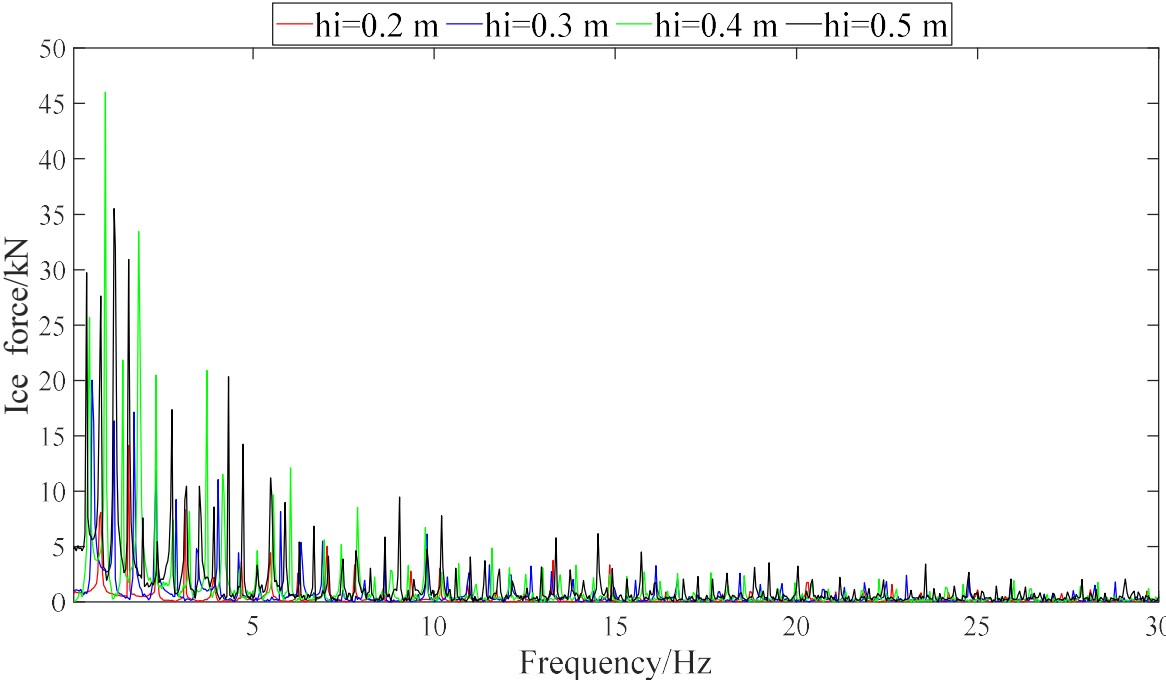

**Figure 17.** Frequency domain curves with different ice thickness (hi).

In Table 7, the average ice force, maximum ice force and variance of ice thickness are presented. The data curves are given in Figure 18. We can know that the average ice force, maximum ice force and variance will increase with the increase of the ice thickness by analyzing the curves under different ice thicknesses. The influence of ice thickness on maximum values are higher than those on average values.

**Table 7.** Statistics of ice force under different Ice thickness (hi).

| Ice Thickness/m | Average Ice Force/kN | Maximum Ice Force/kN | Variance/kN | The Frequencies of the First Four Peaks/Hz |
|---|---|---|---|---|
| 0.2 | 18.13 | 61.9 | $0.22 \times 10^3$ | 0.80/1.56/2.36/3.12 |
| 0.3 | 31.63 | 109.6 | $0.80 \times 10^3$ | 0.56/1.16/1.72/2.32 |
| 0.4 | 46.28 | 204.5 | $2.68 \times 10^3$ | 0.48/0.92/1.40/1.84 |
| 0.5 | 78.31 | 244.5 | $3.14 \times 10^3$ | 0.40/0.80/1.16/1.56 |

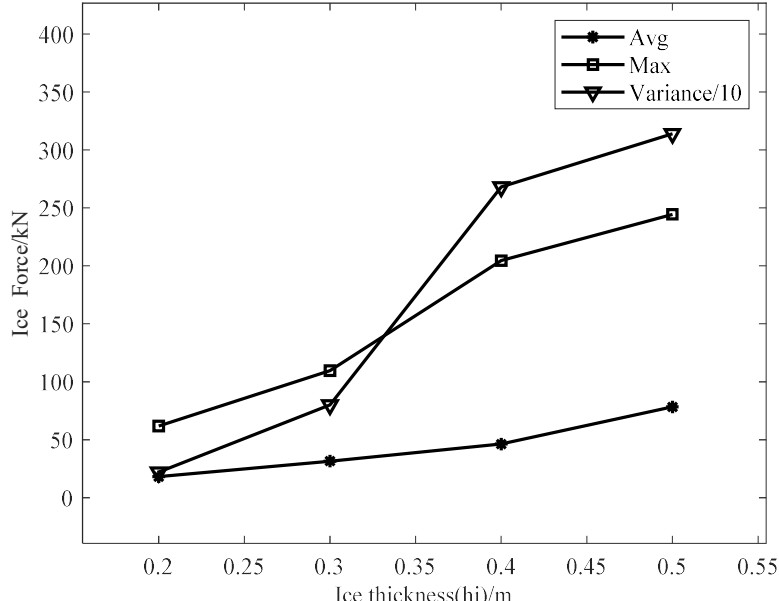

**Figure 18.** Change trend of variance and ice force with ice thickness (hi).

### 4.4. Bending Strength

Bending strength is a factor of ice bearing capacity for each ice grid. It is varying during the ice growth process. Herein, the bending strength used for analysis ranges from 300 to 900 kpa, with an interval of 200 kPa. The simulated time history curves of longitudinal ice force are shown in Figure 19. It can be seen from the time history curves that the peaks of ice load in four cases is quite different. It shows that the bending strength can affect icebreaking efficiency clearly.

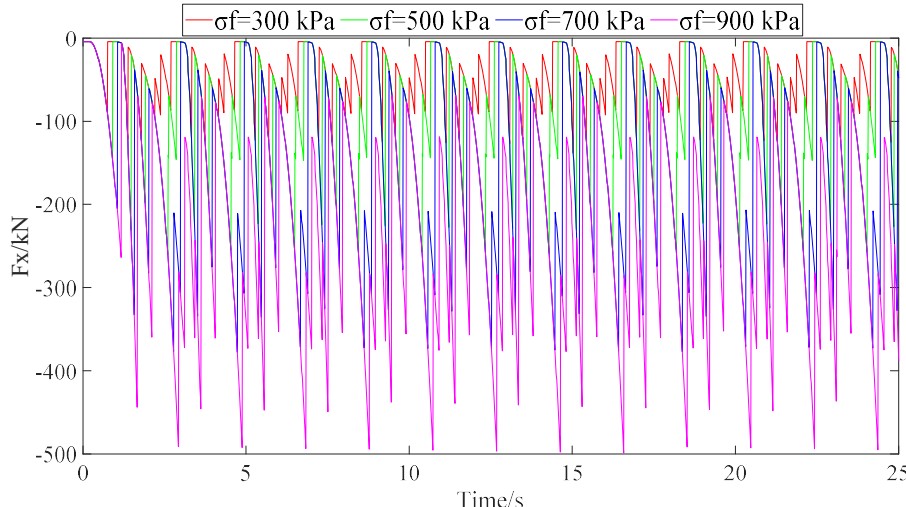

**Figure 19.** Time history curves of total ice force under different bending strength ($\sigma_f$).

Four time series of ice force in Figure 19 are transformed to frequency domain curves by using FFT. The results are shown in Figure 20. It is found that the frequencies of the peaks are all coincident. The frequency corresponding to the peak is a multiple of 0.5 Hz, such as 0.5 Hz, 1.0 Hz and 1.5 Hz, while the block length is 1.002 m and the icebreaking speed is 0.514 m/s. The ratio is about 0.5, which coincides with the frequency of 0.5 Hz when the peak occurs in the frequency domain curves. This shows that the frequency of peaks simulated is highly related to the length of the ice block. The changes in bending strength will not affect the size of the ice block.

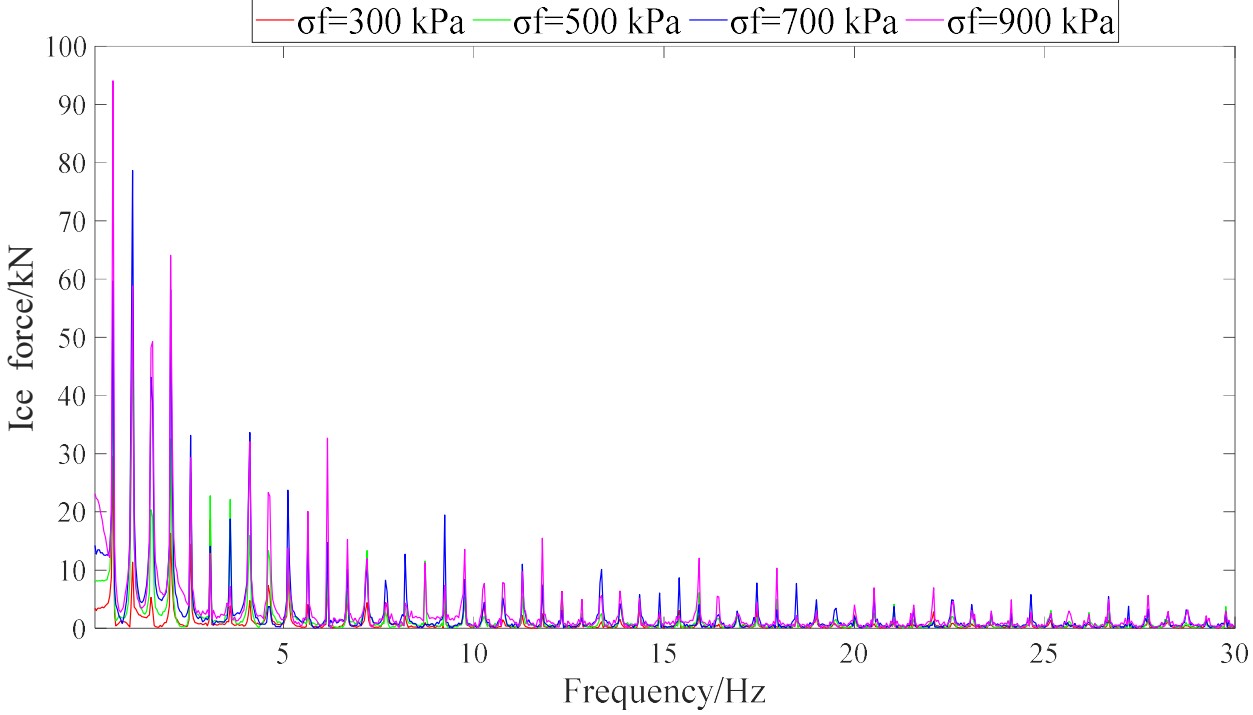

**Figure 20.** Frequency domain curves with different bending strength ($\sigma_f$).

In Table 8, the average ice force, maximum ice force and variance under different bending strengths are presented. The data curves are shown in Figure 21. We can find that the average ice force, maximum ice force and variance will increase with the increase of the bending strength. The average ice force, maximum ice force and variance are greatly affected by the change of the bending strength.

**Table 8.** Statistics of ice force under different bending strength ($\sigma_f$).

| Bending Strength/kPa | Average Ice Force/kN | Maximum Ice Force/kN | Variance/kN | The Frequencies of the First Four Peaks/Hz |
|---|---|---|---|---|
| 300 | 41.98 | 138.1 | $1.01 \times 10^3$ | 0.48/1.00/1.48/2.00 |
| 500 | 90.43 | 260.0 | $4.34 \times 10^3$ | 0.48/1.00/1.48/2.00 |
| 700 | 153.61 | 377.6 | $10.01 \times 10^3$ | 0.48/1.00/1.48/2.00 |
| 900 | 234.62 | 498.8 | $12.90 \times 10^3$ | 0.48/1.00/1.48/2.00 |

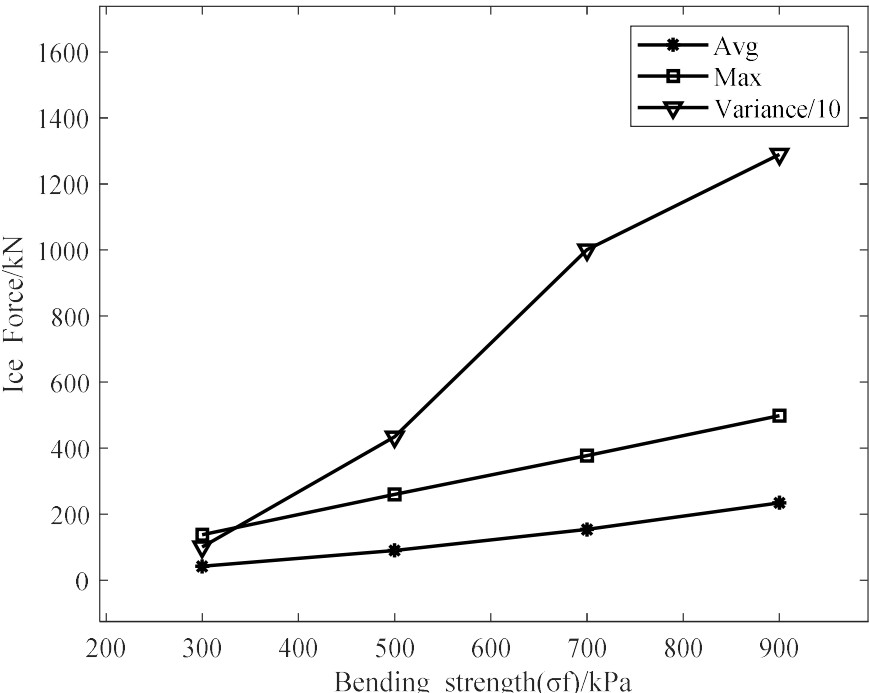

**Figure 21.** Change trend of variance and ice force with bending strength ($\sigma_f$).

### 4.5. Crushing Strength

Crushing strength is used to calculate interaction force on contact surface between ice and structures. It is also important to consider how it will influence the resulting load. The values of 1000 kPa, 3000 kPa, 5000 kPa and 7000 kPa are considered to study the effect on ice force in the simulation, and time history curves of ice force are shown in Figure 22.

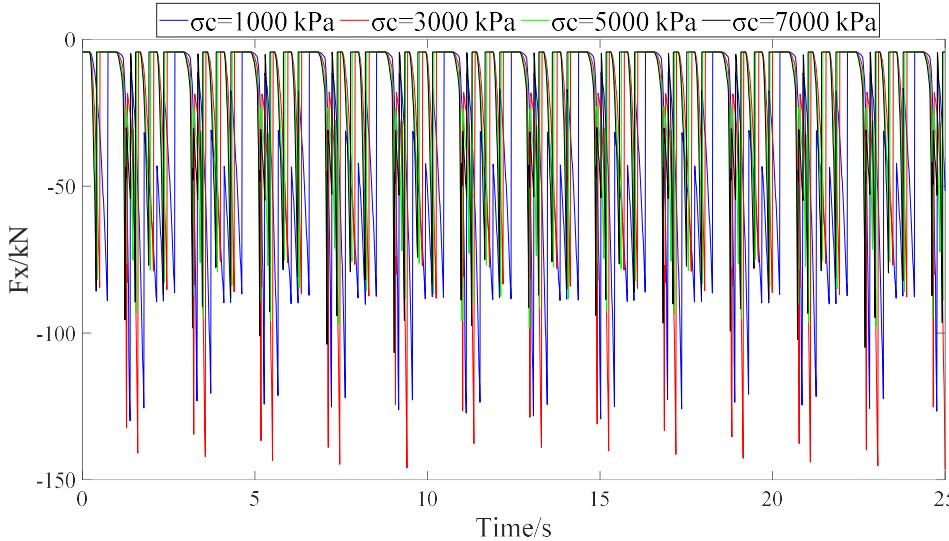

**Figure 22.** Time history curves of total ice force under different crushing strength ($\sigma_c$).

Four time series of ice force in Figure 22 are transformed to frequency domain curves by using FFT. The resulting curves are shown in Figure 23. It can be seen from the figure that large peaks all appear in the low frequency region, and the frequency of peaks is the same under different crushing strength. This shows that the changes in crushing strength will not affect the size of the ice block.

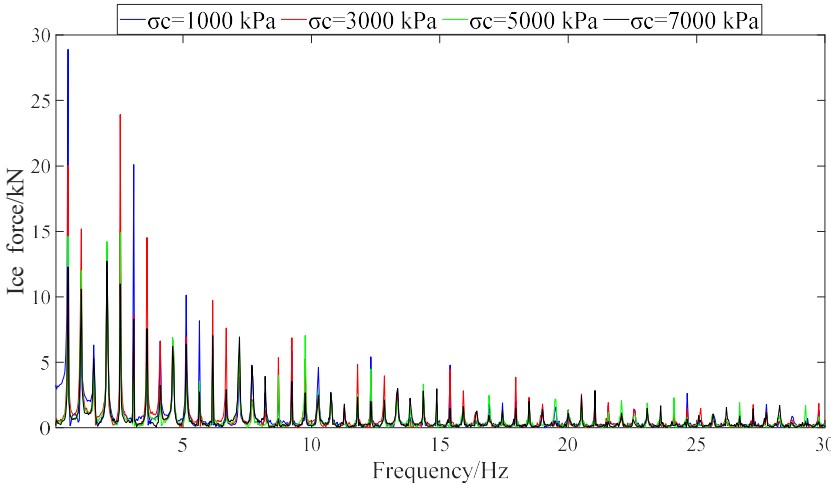

**Figure 23.** Frequency domain curves with different crushing strength ($\sigma_c$).

In Table 9, the average ice force, maximum ice force and variance of crushing strength are presented. The data are plotted in Figure 24. It is found that the average ice force, and variance will decrease with the increase of the crushing strength. The average ice force and crushing strength follow an approximately linear trend. The maximum ice force at 3000 kpa is greater than that of other cases, but there is no clear trend for how maximum values will be affected. The reason is that as ice crushing strength increases, for a single ice–hull interaction event, it will take less time for ice force to reach up to ice bearing load. The ice force will drop to zero subsequently until colliding with next ice grid. The average value and fluctuations of ice load tends to decrease in general.

**Table 9.** Statistics of ice force under different crushing strength ($\sigma_c$).

| Crushing Strength/kPa | Average Ice Force/kN | Maximum Ice Force/kN | Variance/kN | The Frequencies of the First Four Peaks/Hz |
|---|---|---|---|---|
| 1000 | 40.79 | 129.9 | $0.98 \times 10^3$ | 0.48/1.00/1.48/2.00 |
| 3000 | 25.62 | 145.9 | $0.92 \times 10^3$ | 0.48/1.00/1.48/2.00 |
| 5000 | 20.06 | 98.3 | $0.52 \times 10^3$ | 0.48/1.00/1.48/2.00 |
| 7000 | 17.91 | 106.7 | $0.47 \times 10^3$ | 0.48/1.00/1.48/2.00 |

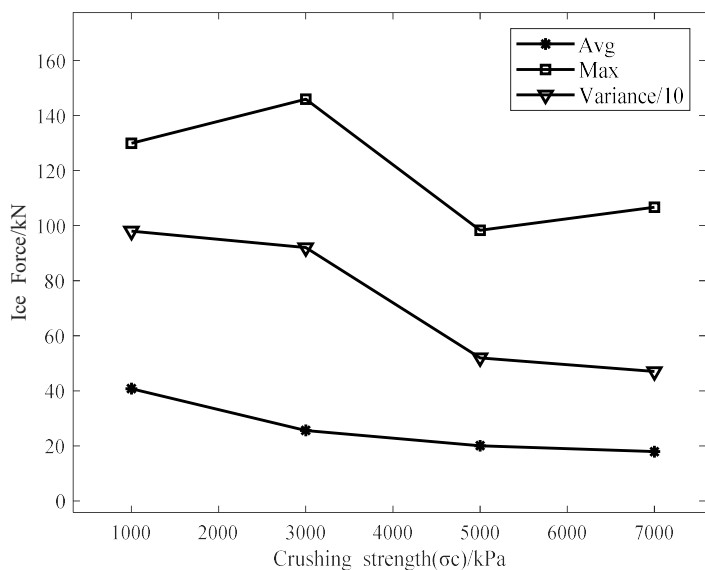

**Figure 24.** Change trend of variance and ice force with crushing strength ($\sigma_c$).

## 4.6. Friction Coefficient

The friction coefficient is the roughness of the contact surface between the hull and the ice. The friction coefficient of the hovercraft is slightly larger than that of the steel ship. Take friction coefficient as 0.15, 0.2, 0.25 and 0.3 to study the effect on ice force in this section, and time history curves of total ice force are shown in Figure 25.

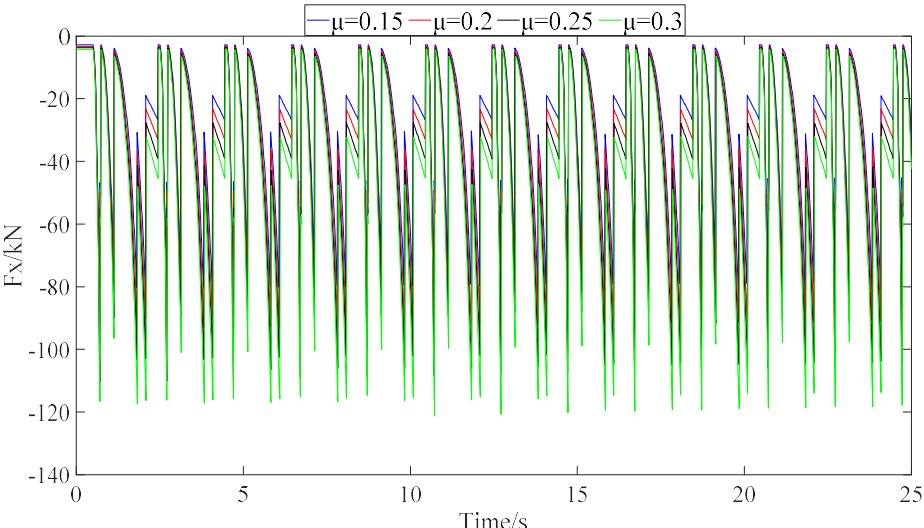

**Figure 25.** Time history curves of total ice force under different friction coefficients (μ).

All time series of ice force in Figure 25 are transformed to frequency domain curves by using FFT. The resulting curves are shown in Figure 26. It can be seen from the figure that the frequency of peaks is the same under different friction coefficients. This shows that the changes in friction coefficient will not affect the size of the ice block. It can be found from the figure that there will be small peaks between the main peaks. This is because the breaking of ice block does not occur at the same time during the contact between the hovercraft node and the ice block.

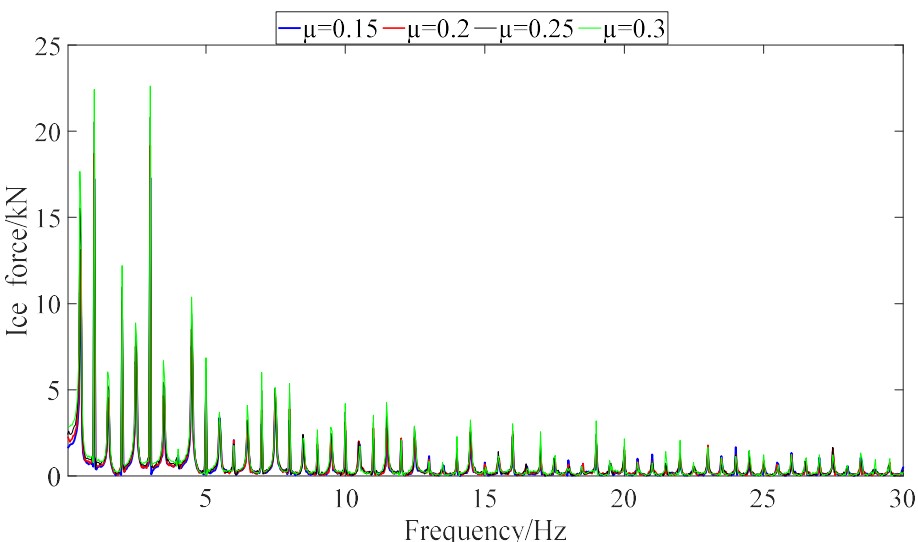

**Figure 26.** Frequency domain curves with different friction coefficients (μ).

In Table 10, the average ice force, maximum ice force and variance of friction coefficient are presented. The curves are shown in Figure 27. We can know that the average ice force, maximum ice force and variance will increase with the increase of the friction coefficient, and the effect of friction coefficient is small. The peaks in the time history curve are in good

agreement, so the friction coefficient has little effect on the icebreaking efficiency. Variance varies greatly with the friction coefficient.

**Table 10.** Statistics of ice force under different friction coefficients (μ).

| Friction Coefficient | Average Ice Force/kN | Maximum Ice Force/kN | Variance/kN | The Frequencies of the First Four Peaks/Hz |
|---|---|---|---|---|
| 0.15 | 29.42 | 92.15 | $0.48 \times 10^3$ | 0.48/1.00/1.48/2.00 |
| 0.20 | 34.02 | 101.3 | $0.61 \times 10^3$ | 0.48/1.00/1.48/2.00 |
| 0.25 | 38.77 | 110.3 | $0.71 \times 10^3$ | 0.48/1.00/1.48/2.00 |
| 0.30 | 43.67 | 121.2 | $0.96 \times 10^3$ | 0.48/1.00/1.48/2.00 |

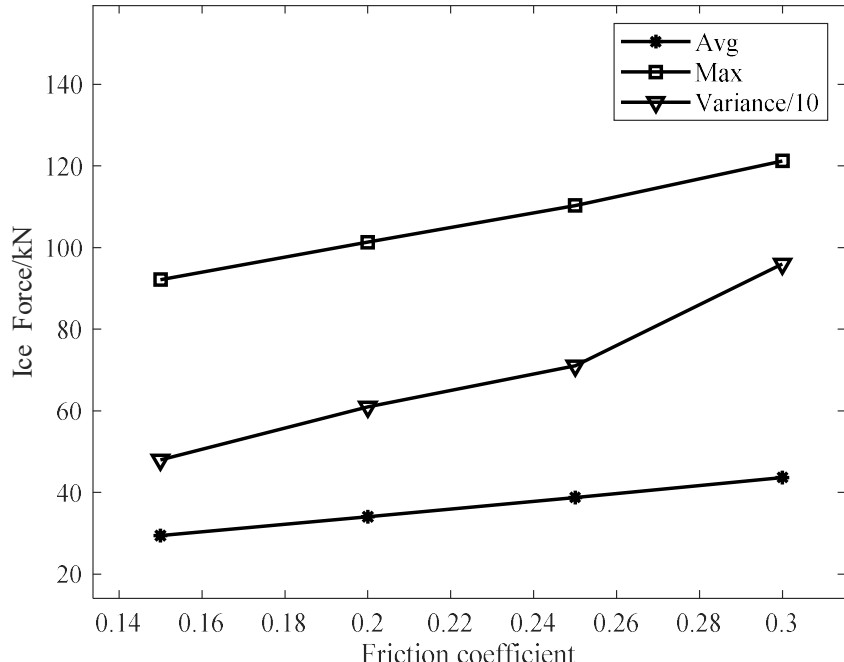

**Figure 27.** Change trend of variance and ice force with friction coefficient (μ).

### 4.7. Young's Modulus

Young's modulus is also an important parameter of sea ice. Take Young's modulus as 1 GPa, 2 GPa, 3 GPa and 4 GPa to study the effect on ice force in this section, and time history curves of total ice force are shown in Figure 28. The time series of ice force in Figure 28 are transformed to frequency domain curves by using FFT. The resulting curves are shown in Figure 29. It is clear from the figure that the frequency of peaks is different under different Young's modulus. Therefore, Young's modulus is also one of the factors that affect the length of the ice block.

In Table 11, the average ice force, maximum ice force and variance of Young's modulus are presented. The curves are shown in Figure 30. We can know that the average ice force will decrease with the increase of the Young's modulus. It can be seen from the time history curves that the ice breaking interval gradually increases as the Young's modulus increases. It shows that the Young's modulus can affect icebreaking efficiency. The change of maximum ice force and variance is relatively random.

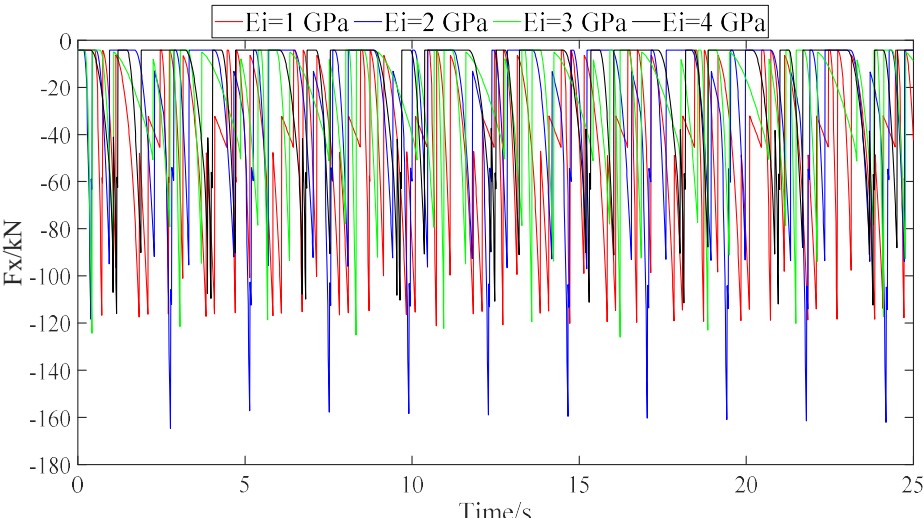

**Figure 28.** Time history curves of total ice force under different Young's modulus ($E_i$).

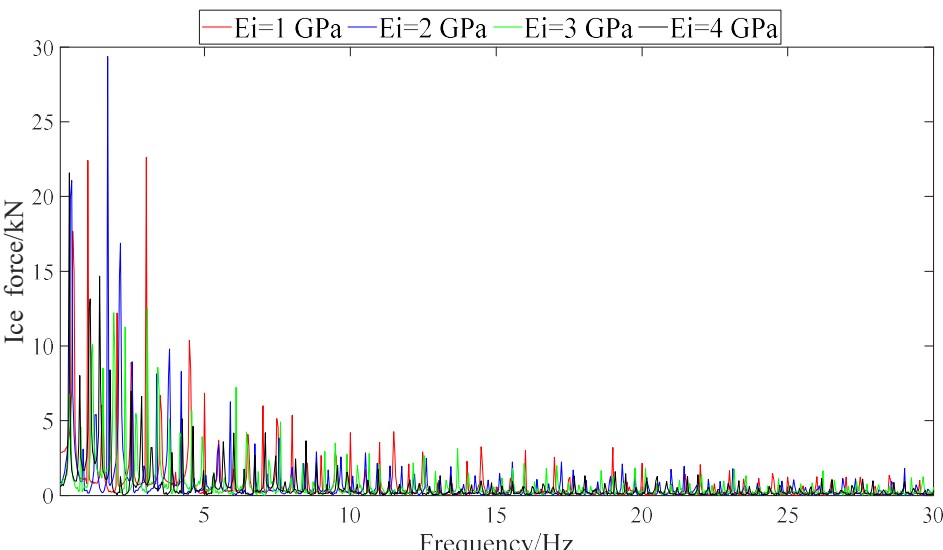

**Figure 29.** Frequency domain curves with different Young's modulus ($E_i$).

**Table 11.** Statistics of ice force under different Young's modulus ($E_i$).

| Young's Modulus/GPa | Average Ice Force/kN | Maximum Ice Force/kN | Variance/kN | The Frequencies of the First Four Peaks/Hz |
|---|---|---|---|---|
| 1 | 43.67 | 121.2 | $0.96 \times 10^3$ | 0.48/1.00/1.48/2.00 |
| 2 | 28.30 | 164.6 | $1.05 \times 10^3$ | 0.44/0.84/1.28/1.68 |
| 3 | 26.74 | 125.1 | $0.55 \times 10^3$ | 0.36/0.76/1.16/1.52 |
| 4 | 18.41 | 115.9 | $0.58 \times 10^3$ | 0.36/0.72/1.08/1.40 |

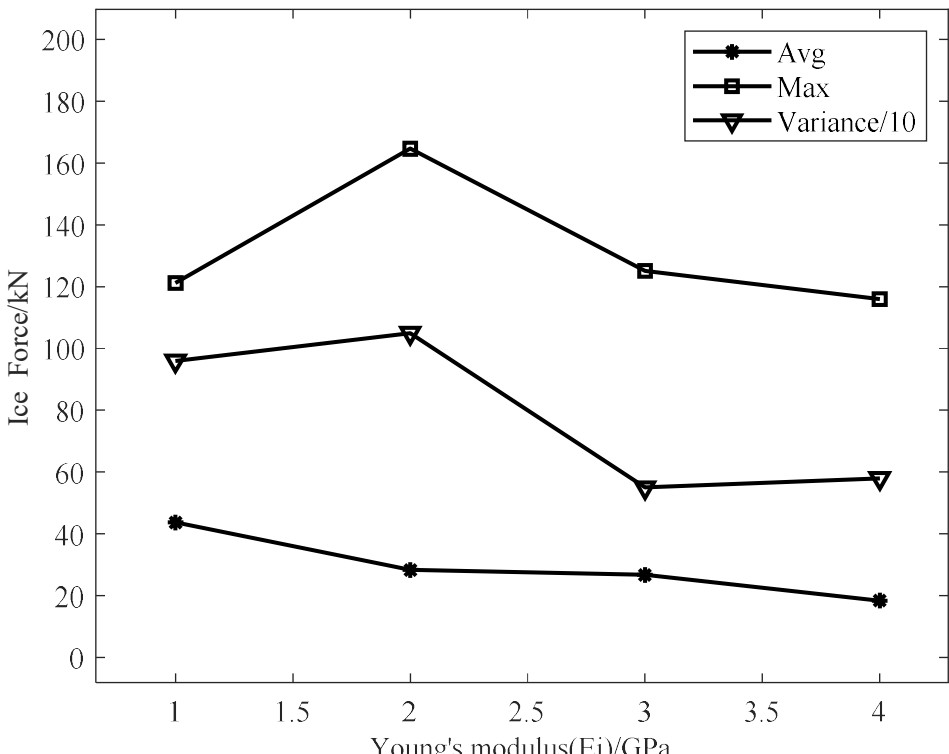

**Figure 30.** Change trend of variance and ice force with Young's modulus ($E_i$).

## 5. Discussion

The case 3 in Table 4 is selected as the basic case. The statistics of ice loads under different factors aforementioned in Section 4 are compared with the basic case and expressed by means of dimensionless analysis. The corresponding average, maximum and variance value are presented in Figure 31.

It can be found from Figure 31 that the influence of bending strength, ice thickness, fracture coefficient and friction coefficient on the average ice force is positively correlated, and the sensitivity decreases sequentially. The length coefficient, crushing strength and Young's modulus are negatively correlated with the average ice force, and the sensitivity decreases sequentially. The length coefficient gives the largest negative effect on ice load non-monotonously. This is because interval of subsequent ice failures increases and unloading duration of ice with hull increases as the length coefficient increases. The maximum ice force is also the most sensitive to ice thickness and bending strength since the bearing capacity of each ice grid is highly dependent on ice thickness and bending strength. It can be seen from Figure 31c that the bending strength has a great influence on the variance, which increases approximately exponentially. Although other variables also have an impact on the variance, the influence trend tends to be flat. Among all factors considered herein, the ice thickness exposes the most influence on ice load from the simulation by using the present method.

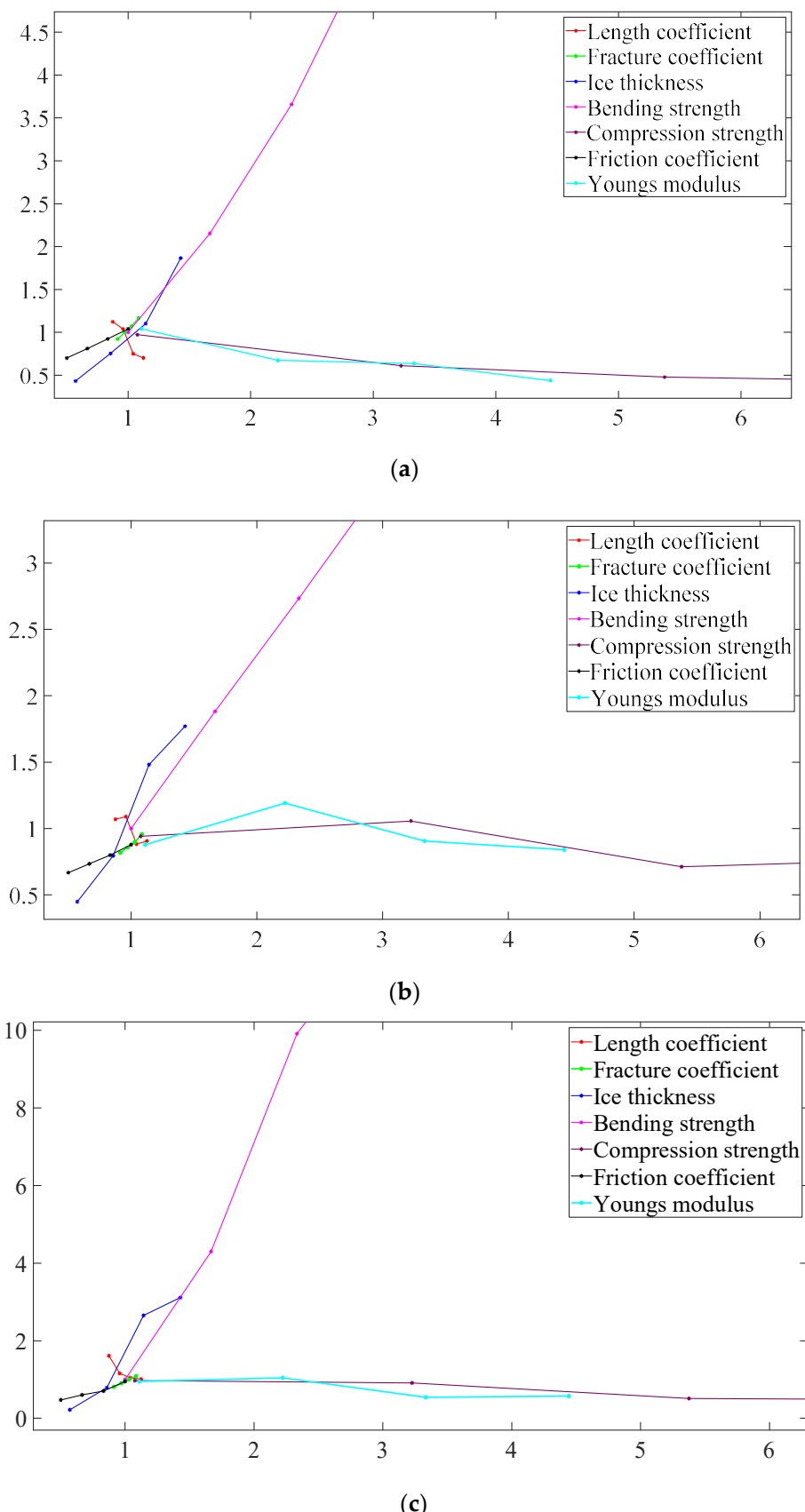

**Figure 31.** Dimensionless ice force compared to the basic case. (**a**) Average ice force; (**b**) Maximum ice force; (**c**) Variance.

## 6. Conclusions

This paper analyzes the ice force acting on the hovercraft during low-speed icebreaking, and studies the icebreaking mechanism of the hovercraft and the failure pattern of sea ice under ideal assumptions. Based on the phenomenon of circumferential crack, combined with the idea of discretization, the hovercraft and ice are discretized in the simulation.

The numerical model of icebreaking of hovercraft at low speed was established by these methods, and the numerical simulation study was carried out and validated with three cases in the ice basin model tests. It shows that the present method gives a reasonable prediction on ice load exposed to a hovercraft. We have carried out numerical simulation calculations under three cases through numerical simulation methods. The error between the average resistance result and that of the model test in case 1 is 29.58%, the error in case 2 is 8.15% and the error in case 3 is 6.09. The error between the maximum resistance and that of the model test in case 1 is 66.7%, the error in case 2 is 26.9% and the error in case 3 is 35.8.

Parametric analysis is also tried to study the influence of different factors on ice load. It is found that the average ice force increases with the increase of ice thickness, bending strength, fracture coefficient and friction coefficient, and decreases with the increase of length coefficient, Young's modulus and crushing strength. By using dimensionless analysis, it is concluded that the ice thickness shows the greatest influence on the ice force. It is noted that gas leakage when the ship first contacts with ice, which has not been considered in this numerical simulation for the time being, and will be studied in the future.

**Author Contributions:** Conception, L.Z.; data analysis, J.J. and Y.G.; data interpretation, L.Z.; data collection, S.D. and Y.G.; literature search, S.D.; writing, J.J. All authors have read and agreed to the published version of the manuscript.

**Funding:** This research was funded by the National Natural Science Foundation of China, grant number 51809124, 51911530156.

**Institutional Review Board Statement:** Not Applicable.

**Informed Consent Statement:** Not Applicable.

**Data Availability Statement:** Not Applicable.

**Acknowledgments:** This research is funded by the National Natural Science Foundation of China, grant number 51809124, 51911530156.

**Conflicts of Interest:** The authors declare no conflict of interest.

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
