# Peer review of "Numerical Simulation of the Ice Breaking Process for Hovercraft"

_jmse, doi:10.3390/jmse9090928_

Round 1

Reviewer 1 Report

Sensitivity analysis was performed on the main parameters used for ice resistance calculation using a numerical tool and applied to a hovercraft ship in this paper. This paper compared the results of the model test published as a reference 14. The originality is insufficient, but the engineering application is not bad.

Symbols used in the figure need explanations (some symbols are missing)

The part where the symbol in the formula and the symbol in the text do not match needs to be corrected.

Some symbols used in formulas are not matched. (It that need to be modified with the same symbols in equations (3) to (5).

It need to insert reference number in line number 132, 150, 161, 187

The reference number of line 181 have to check. Formula (17) ~ (19) is described in reference 8. I can’t find reference 12.

The unit on line 199 is not correct.

Authors compared the items of in each section with the length of the ice block for sensitivity analysis in section 4.3, 4.4, 4.5, 4.6 but there is no explanation for the correlation between the size of ice block and the ice force, so the logic for the explanation is insufficient.

In particular, there is no need to explain the relationship between crushing stress and size of ice block in 4.6. (line 444, 450)

Author Response

Response to Reviewer 1

Sensitivity analysis was performed on the main parameters used for ice resistance calculation using a numerical tool and applied to a hovercraft ship in this paper. This paper compared the results of the model test published as a reference 14. The originality is insufficient, but the engineering application is not bad.

Response: Thanks for reviewer’s valuable comments with great deep insight into the proposed method. We consider the value of engineering application. We adopted the simplified circumferential icebreaking approach and got not bad results. The present study is significant to preliminary design of new icebreaking hovercraft, which assists operation possibility for existing hovercraft.

Symbols used in the figure need explanations (some symbols are missing)

Response: Yes. Thank you for this good suggestion. We have revised Figure 3 and the symbols on page 3, line 85-88.

The part where the symbol in the formula and the symbol in the text do not match needs to be corrected.

Response: Yes. Thank you for this good suggestion. We checked and revised the symbol in the formula and the symbol in the text do not match carefully.

Some symbols used in formulas are not matched. (It that need to be modified with the same symbols in equations (3) to (5).

Response: Maybe you mean the symbol  in equations (17) to (18) do not match the symbol  in equations (3) to (5). Where  is the forward component of . The parameter  is the component of  along to the contact surface in the vertical plane. The parameter  is the normal component of  along the hull contact nodes.

It need to insert reference number in line number 132, 150, 161, 187

Response: Yes. Thank you for this good suggestion. We have inserted reference number in line number 132, 150, 161, 187.

The reference number of line 181 have to check. Formula (17) ~ (19) is described in reference 8. I can’t find reference 12.

Response: Yes. This is a good suggestion. We have corrected and cited it as reference. And we deleted reference [12].

The unit on line 199 is not correct.

Response: Yes. This is a good suggestion. We have revised the unit on line 199.

Authors compared the items of in each section with the length of the ice block for sensitivity analysis in section 4.3, 4.4, 4.5, 4.6 but there is no explanation for the correlation between the size of ice block and the ice force, so the logic for the explanation is insufficient.

Response: The size of the ice block is determined by equation (11), and is only related to the length coefficient, ice thickness and Young’s modulus. Therefore, there is no direct relationship between ice power and ice size. And we have also made a revision to section 4.2.

In particular, there is no need to explain the relationship between crushing stress and size of ice block in 4.6. (line 444, 450)

Response: Yes. Thank you for this good suggestion. This is a mistake due to the carelessness of authors. Because in the four cases in this section, only the friction coefficient is different. We have revised it in Section 4.6 (line 453, 459).

Reviewer 2 Report

Reviewers' Comments:

Page 2, lines 74-82: The authors determined the crushing force during hover apron and ice interaction based on contact area. However, the case of contact between them is not clear. In particular, the Figure 2 shows the only one case of contact interface. Please clarify the contact surface corresponding to the thickness of ice and penetration depth in this study.

Page 5, lines 188-189, Equation (20): This formulation is given by Lindqvist (1989). This reference is missing in the current manuscript (“Lindqvist, G. A straightforward method for calculation of ice resistance of ships. In Proceedings of the International Conference on Port and Ocean Engineering under Arctic Conditions (POAC), Lulea, Sweden, 12–16 June 1989; pp. 722–735.).

Page 5, line 199: “320.0m2” to “320.0m2

Page 7, Table 3: Please check the value of ice density and water density. I think the two values are switched. 

Page 9, lines 263-274: These paragraphs are simply describing the data. At least more in-depth findings should be presented.

Page 9, Lines 279-280: “The second reason may be… exist some uncertainties.” This is a very significant statement which is not explained in the current manuscript. Please clarify the experimental uncertainty for ice tank tests in this study.

In Tables 5 to 11, what is the corresponding frequency of each of the parameters? These information could be provided in table. This could help reader understand the significance of the parameters.

Page 18, Lines 443-447: “This shows that the changes… hovercraft node and the ice block.” It is not clear what the relationship between the crushing strength and the ice block size under the friction coefficient. Please clarify this statement.

Author Response

Response to Reviewer 2

Page 2, lines 74-82: The authors determined the crushing force during hover apron and ice interaction based on contact area. However, the case of contact between them is not clear. In particular, the Figure 2 shows the only one case of contact interface. Please clarify the contact surface corresponding to the thickness of ice and penetration depth in this study.

Response: Yes. Thank you for this good suggestion. When hovercraft breaks level ice in polar area, as the hovercraft moves forward, the contact area between ice and the hull increases. This implies that two cases must be considered. We have revised Figure 2 and added another situation. Specifically, as shown in reference [8].

Page 5, lines 188-189, Equation (20): This formulation is given by Lindqvist (1989). This reference is missing in the current manuscript (“Lindqvist, G. A straightforward method for calculation of ice resistance of ships. In Proceedings of the International Conference on Port and Ocean Engineering under Arctic Conditions (POAC), Lulea, Sweden, 12–16 June 1989; pp. 722–735.).

Response: Yes. This is a good suggestion. We have corrected and cited it as reference.

Page 5, line 199: “320.0m2” to “320.0m2”

Response: Yes. This is a good suggestion. We have revised the unit on line 199.

Page 7, Table 3: Please check the value of ice density and water density. I think the two values are switched.

Response: Yes, Thank you for this good suggestion. We have corrected the value of ice density and water density.

Page 9, lines 263-274: These paragraphs are simply describing the data. At least more in-depth findings should be presented.

Response: Yes. Thank you for this good suggestion. Combined the phenomena observed in the experiment, we analyzed the time history curve. And we analyzed the reasons for the phenomenon of force loading and unloading. Specifically, as shown on the page 9, lines 274-280.

Page 9, Lines 279-280: “The second reason may be… exist some uncertainties.” This is a very significant statement which is not explained in the current manuscript. Please clarify the experimental uncertainty for ice tank tests in this study.

Response: Yes. This is a good suggestion. We put forward two possible reasons that will affect the experimental data. Specifically, as shown on the page 9, lines 286-287.

In Tables 5 to 11, what is the corresponding frequency of each of the parameters? These information could be provided in table. This could help reader understand the significance of the parameters.

Response: Yes. Thank you for this good suggestion. We have revised tables 5 to 11. We have added the frequencies of the first four peaks in the table 5 to 11. Hope to help readers understand the significance of parameters.

Page 18, Lines 443-447: “This shows that the changes… hovercraft node and the ice block.” It is not clear what the relationship between the crushing strength and the ice block size under the friction coefficient. Please clarify this statement.

Response: Yes. Thank you for this good suggestion. This is a mistake due to the carelessness of authors. Because in the four cases in this section, only the friction coefficient is different. And we have revised it in section 4.6 (line 453, 459).

Round 2

Reviewer 1 Report

Please find attached file.

Author Response

Manuscript ID: jmse-1335937

Title: Numerical Simulation of Ice Breaking Process for Hovercraft

Dear Editor and Reviewers,

The authors would like to thank the time and great effort you have spent in reviewing our manuscript. We revised the manuscript in accordance with your comments carefully and seriously based on your valuable comments.

According to the comments, we have studied comments carefully point by point and have made correction which we hope to meet with approval.

Revised portion are marked in green in the revised manuscript. The main corrections in the paper and the responds to the reviewer’s comments are as flowing:

Reviewer 1 (Round 2):

?, ?, ?? is still missing the explanation in Fig.1

Response: Yes. Thank you for this good suggestion. We have explained ?, ?, ?? on page 2, line 71 to 73.

Some of symbol is not explained. Please check one by one.

And the crushing force => And the crushing force (???) Line 78.

Response: Yes. This is a good suggestion. We have checked and explained the symbols in line number 80 (page 2), 103 (page 3), 108 (page 3), 177 (page 5), 179 (page 5), 197 (page 5), 198 (page 5).

Please check line 107, 119 and formula (7), (9) etc.

?′ and ?′ are different and give some confusion to the reader.

Response: Yes. Thank you for this good suggestion. We have checked and revised the symbols in line number 107 and 119.

It’s Ok but line 161 looks like [8].

Response: Su introduced the source of formula (13) in his paper. It’s cited from [9], shown as

Please check the symbol in line 186.

Response: Yes. Thank you for this good suggestion. We have checked and revised the symbols in line number 190 (page 5).

The language was checked by a native English-speaking colleague.

Thank you very much!